# The Dynamic Changes of Gut Microbiota during the Perinatal Period in Sows

**DOI:** 10.3390/ani10122254

**Published:** 2020-11-30

**Authors:** Lan Sun, Yu Zhang, Wei Chen, Tian Lan, Yan Wang, Yinbao Wu, Xindi Liao, Jiandui Mi

**Affiliations:** 1National Engineering Research Center for Breeding Swine Industry, College of Animal Science, South China Agricultural University, Guangzhou 510642, China; lansun@163.com (L.S.); zy1019@stu.scau.edu.cn (Y.Z.); weichen@163.com (W.C.); lantian_hn@163.com (T.L.); ywang@scau.edu.cn (Y.W.); wuyinbao@scau.edu.cn (Y.W.); 2Ministry of Agriculture Key Laboratory of Tropical Agricultural Environment, South China Agricultural University, Guangzhou 510642, China; 3Guangdong Provincial Key Lab of Agro-Animal Genomics and Molecular Breeding and Key Lab of Chicken Genetics, Breeding and Reproduction, Ministry of Agriculture, Guangzhou 510642, China

**Keywords:** sow, gut microbiota, perinatal period, dynamic changes

## Abstract

**Simple Summary:**

The gut microbiota in sows is important for the health of the host, and potential benefits may also be transferred to piglets during pregnancy. Therefore, systematic studies investigating the changes in the gut microbiota of sows are needed to elucidate the microbial compositions and functions. This study was conducted at 12 time points to investigate the temporal variations in the gut microbiota of sows using 16S rRNA sequencing. The results provided important information for the gut microbiota of sows during different perinatal periods.

**Abstract:**

The gut microbiota in sows is important for the health of the host, and potential benefits may also be transferred to piglets during pregnancy. Therefore, systematic studies investigating the changes in the gut microbiota of sows are needed to elucidate the microbial compositions and functions. This study was conducted at 12 time points to investigate the temporal variations in gut microbiota on Days 27, 46, 64, 81, 100, and 113 during gestation (G) and Days 3, 5, 7, 10, 14, and 21 during lactation (L). Results suggested that the gut microbiota changed across the perinatal period with microbial function and abundance varying between the prenatal and postnatal periods. The alpha diversity was higher in the postnatal period than in the prenatal period. Thirty-eight genera were distributed between the two periods with *Methanobrevibacter*, *Desulfovibrio*, *Akkermansia*, and *Turicibacter* being enriched in the prenatal period while *Eubacterium*, *Actinobacillus*, *Paludibacter*, *Butyricimonas*, *Megasphaera*, *Succiniclasticum*, *Acidaminococcus*, *and Rummeliibacillus* were enriched in the postnatal period. Analysis done at the different time points of the prenatal period suggested that Days 27 and 113 had more microbial biomarkers than other days. *Bacteroidales*, *Bacteroidia*, and *Prevotella* were enriched on the 27th day, while bacteria belonging to the *Clostridium* and *Ruminococcaceae* were enriched on the 113th day. On the other hand, *Clostridiales*, *Ruminococcaceae*, *Clostridia*, and unclassified *Christensenellaceae* were enriched three days after delivery. Predicted microbial KO functions were also more enriched on Day 27 of the gestation period and Day 3 of the lactation period. Random forest, a machine learning method, was used to identify the top five important genera of *Megasphaera*, *Stenotrophomonas*, *Phyllobacterium*, *Catenibacterium*, and *Turicibacter*, while the most important function was arginine and proline metabolism. These systematic results provide important information for the gut microbiota of sows.

## 1. Introduction

Several trillion microbial cells present in the gut play different roles in the gut health of adults, and their cooperation and competition also affect disease development [1]. It was found that intestinal microbes in women undergo significant changes during pregnancy [2], and this change not only affects the health of the mother, but also affects the health of the fetus [3,4]. Similarities in the genetic background between pigs and humans indicates that human and pig experimental results can be shared between them [4]. For pig farmers, sows are very important for productivity since each sow with one parity can produce more than 10 piglets. Therefore, maintaining the health of sows is a very important mission for improving the productivity of pig farms. Studies conducted on pregnant sows in recent years have indicated that the pregnancy status of sows not only affects overall productivity, but also affects the growth of piglets [3,4]. However, most studies have only focused on the immunity and physiological effects of the sow on the piglets [3,4]. The role of the sow’s gut microbiota on the health of the sow and piglet has rarely been investigated.

The gut microbiota in the sow ferments carbohydrate and protein residues, resulting in metabolic substrates such as short fatty acids and aryl hydrocarbon compounds, which provide nutrition for the host and also stimulate the establishment and improvement of immunity [5,6]. These substrates and microbiota can also be transferred or inherited by the piglets, thus playing an important role in the development and health of the piglets. Pregnancy was also been found to be associated with a shift in the composition and function of gut microbiota in sows [7]. Therefore, longitudinal studies are required to investigate the changes in the gut microbiome of sows throughout pregnancy and during lactation. In addition, manipulating the ecological balance of the gut microbiota can be an important method for improving the health of pregnant sows because there is restricted feeding during late pregnancy. Therefore, elucidating the dynamics of gut microbiota from pregnancy to the end of lactation can provide basic information that can help in manipulating the microbiota of sows in the future.

A study showed that the proper addition of inulin can improve the metabolic and inflammatory status of pregnant women [8]. In addition, 16S rRNA analysis of the microbiota at four time points before delivery found that the intestinal microbial community of sows changed after the addition of inulin [8]. Analysis done at four time points before and after delivery found that sows have metabolic syndrome during the perinatal period, which influences the rapid changes of the intestinal microbiota [9]. Studies on the microbiota at each of the time points during pregnancy also showed that the microbiota changes continuously during pregnancy. The longer the gestational age, the lower the abundance of colonic bacterial species is [10]. However, information about gut microbiota changes throughout the reproductive cycle in pregnant sows is limited. Therefore, we designed this systematic study to illustrate the dynamical changes of gut microbiota in sows throughout the reproductive cycle. Data obtained from this study will provide comprehensive information for exploring microbial changes throughout the perinatal period.

We chose six time points, both before and after birth (total 12 time points), in the study where the next-generation sequencing technology using 16S rRNA with the culture-independent method was used to investigate the periodic changes in intestinal microbes by comparing the perinatal stages of microbes in pregnant sows. This study also aimed at comprehensively supplementing the types and changes of microorganisms at various stages of pregnancy through analysis and comparison.

## 2. Materials and Methods

### 2.1. Sample Collection

All Large-White × Landrace sows used in the study were housed under the same breeding conditions in a commercial farm in Guangdong Province, China. Samples were collected from April to May 2017 with the sows selected for sample collection all being healthy. Pregnant sows with a similar body weight and health were randomly selected from different animal houses at the same pig farm at G27 (n = 11), G46 (n = 11), G64 (n = 8), G81 (n = 9), and G100 (n = 9) days of gestation. A total of 23 adult sows (average parity = 3 to 5) were also selected and concurrently delivered to continuously track the dynamic development of the gut microbiota from G113 days of gestation to 21 days of lactation. To ensure simultaneous delivery, thirty candidate healthy sows were intramuscularly injected with cloprostenol (0.2 mg per sow) at 9:00 a.m. on the 113th day of pregnancy. Sows that had not delivered after three hours of injection were excluded from the study. Fecal samples were collected at 113 (n = 13) days of gestation and L3 (n = 2), L5 (n = 19), L7 (n = 20), L10 (n = 13), L14 (n = 23), and L21 (n = 22) days of lactation. To avoid the influence of time on the results, all fresh feces were collected between 9:00 and 10:00 a.m. Samples were collected from fresh excrement where a sterile stainless-steel spoon was used to collect the sample from the center of the feces to avoid environmental contamination. The samples were then transferred to three sterile tubes followed by immediate placement in liquid nitrogen. Samples were then transported to the laboratory and stored at −80 °C for analysis. Because during parturition, the sow decreased feed intake and increased energy consumption and sows started to excrete three days after delivery, we only collected a few samples on Day 3 of lactation. In total, we collected 160 samples from fresh feces. All sows were fed on the same diet during the perinatal period, which was primarily based on corn and soybean meal and was designed to meet or exceed the energy requirements of National Research Council (NRC) Nutrient Requirements (2012). There was free access to water and feed for all the sows. No antibiotics were used on the sows during the sampling period of the study.

### 2.2. DNA Extraction, 16S rRNA Amplification, and Illumina MiSeq Sequencing

Microbial DNA was extracted by sampling 300 mg of feces from each sample. Genomic DNA was isolated by adding bead-beating using a QIAamp Power Fecal DNA Kit (Qiagen, Hilden, Germany) following the manufacturer’s instructions. Zirconium glass beads (400 mg; 0.1 mm diameter, BioSpec Products, Bartlesville, OK, USA) were added to the extraction system followed by vigorous vortexing (twice) using a FastPrep-24 Instrument (MP Biomedicals, Solon, OH, USA) at a speed of 6.0 m/s for 90 s. Extracted DNA was confirmed using agarose gel electrophoresis.

The 341F (5′-ACTCCTACGGGAGGCAGCAG-3′) forward primer and 806R (5′-GGACTACHVGGGTWTCTAAT-3′) reverse primer were used to amplify the V3-V4 hyper-variable region of the 16S rRNA gene. PCR conditions were: pre-denaturation at 94 °C for 4 min, denaturation at 94 °C for 30 s, annealing at 50 °C for 45 s, and elongation at 72 °C for 30 s for 25 cycles. Finally, extension was done at 72 °C for 5 min. The PCR product was purified and used to construct a library, and then, paired-end sequencing (2 × 250) was performed on a MiSeq platform (Illumina, San Diego, CA, USA) from Novogene Co. Ltd. (Beijing, China).

### 2.3. Sequence Analyses

To obtain accurate and reliable results during bioinformatics analysis, the Quantitative Insights into Microbial Ecology (QIIME) program (v1.9.1) was used to filter raw data [11]. The fragments were then clipped and deleted to a <25 mass score and >225 bp length, respectively. Low-quality sequences were removed, and the remaining sequences were processed and analyzed using the QIIME (v1.9.1) software. The software was used to cluster data into species taxonomic units operational taxonomic units (OTUs) with 97% similarity, and chimeras and error sequences were removed from the optimized data [12]. A genus-level phylogenetic tree was constructed using QIIME’s (v1.9.1) built-in scripts and imaged using R (v3.0.3) software. The Shannon index was used to calculate the observed OTU, while the Chao index was used to evaluate alpha diversity using phyloseq and plot sparse curves using the R (v3.0.3) software. OTU-based Bray–Curtis and weighted and un-weighted UniFrac principal coordinate analysis (PCoA) were then performed using the QIIME (v1.9.1) software. The Log2Fold change of phyloseq in R was used to perform the differential analysis of perinatal intestinal microbes, especially to analyze the microbial changes before and after birth (q < 0.05). The machine learning method random forest was used to identify the top 29 genera and their functions during the perinatal period.

### 2.4. Prediction of Molecular Functions Based on 16S rRNA Data Using PICRUSt

The Phylogenetic Investigation of Communities by Reconstruction of Unobserved States (PICRUSt) was predicted using the 16S rRNA-tagged gene sequence [13]. Phylogenetic studies were performed with regards to existing complete genomic sequences. OTUs were selected for the closed reference method of the Greengenes 13_8 reference database using the pick_closed_reference_otus.py script package of the QIIME software. The OTU table was then normalized based on the copy number of the 16S rRNA gene, and the metagenomics were predicted from the Kyoto Gene and Genomic Encyclopedia (KEGG) catalog. The LEfSe software was used to calculate the difference of the microbial composition and functions with the default LDA score being set to 2.0 and the Kruskal–Wallis and Wilcoxon detection significance thresholds set at 0.05 [14].

## 3. Results

### 3.1. Fecal Microbiological Structure during the Perinatal Period in Sows

Twelve time points were selected throughout the perinatal period in sows to illustrate the microbial structure of feces during the perinatal period including prenatal (27, 46, 64, 81, 100, and 113 days) and postpartum (3, 5, 7, 10, 14, and 21 days). A total of 160 fecal samples were collected, but only 147 samples were analyzed because samples with quality control failures and those with significant differences in the unweighted distance were excluded from the classification. After quality control, a total of 12,987,510 high-quality sequences were generated from the stool samples with an average of 81,171 (range 44,859–93,276) high-quality sequences per sample. The sparse curve of 147 samples with a minimum cutoff of 97% sequence identity was nearly steady, suggesting that the sampling depth was sufficient to characterize the microbiota in the fecal samples.

The abundance and diversity of the stage-producing microorganisms throughout the perinatal period of sows were determined by calculating the Chao 1, Shannon, and observed OTU indices. The results suggested that the Chao 1, Shannon, and observed out indices presented at different perinatal stages were significantly affected (*p* < 0.01, *p* < 0.01, and *p* < 0.05, respectively). The three indices changed in the pregnancy stage where they all increased significantly from the pre-pregnancy to post-pregnancy stage (Figure 1A–C). Non-metric multidimensional scaling (NMDS) was used to identify the differences of the gut microbiota in the sows over a period of time. The obtained results suggested that microbial species are separated significantly during the pre- and post-pregnancy stages (Figure 1D). The NMDS of the pre- and post-pregnancy gut microbiota in sows was identified and used to clarify the effects of time on the microbiota during the pre- and post-pregnancy periods. The obtained NMDS graph shows that the distribution of prenatal microbial species changes over time with the late time of the prenatal period (G100 and G113) being significantly separated from the early stages (Figure 1E). After delivery, gut microbiota in sows was clustered into three groups: 3 and 5 days; 7, 10, and 14 days; and 21 days (Figure 1F). These results suggest that gut microbiota in sows is mainly influenced by pre- and post-pregnancy stages and differences in gut microbiota exist over time before and after delivery.

Fourteen phyla were identified from the entire perinatal gut microbiome in sows based on identification using the 97% sequence identity of 16S rRNA gene sequences (Figure 2A). Firmicutes and Bacteroidetes were the most abundant bacterial phyla in the feces with the average abundance of both in all samples being 83.8%. In addition, the Tenericutes, Spirochaetes, Proteobacteria, and Actinobacteria mean relative abundances were greater than 1% of the total sequences, and so, they were also regarded as predominant bacterial phyla. All these major bacterial phyla accounted for 98.1% of the total sequence in the fecal samples. The remaining bacterial phyla including Fibrobacteres, Cyanobacteria, Euryarchaeota, Lentisphaerae, Verrucomicrobia, Planctomycetes, Fusobacteria, and TM7 were considered as being low abundance bacterial phyla since they had sequence frequencies accounting for <1% of the total sequences and only made up 1.9% of the total sequence in all fecal samples.

In total, ninety-three taxa were observed in the fecal microbial community at the genus level, though an average of 67% in all sequences were not identified (Figure 2B). A total of 17 of the most dominant bacterial populations defined as having a relative abundance of more than 1% of the total sequence are presented for the sake of clarity and visualization. These 17 major bacterial taxa accounted for more than 80.21% of the total sow fecal microbiota sequence. The unclassified-*Ruminococcaceae* (average relative abundance: 20.16%) belongs to the Firmicutes phylum and was the highest taxonomic group in the sow fecal community. Classified taxa in the Firmicutes phylum at the family level were *Christensenellaceae*, *Lachnospiraceae*, *Clostridiaceae*, *and Mogibacteriaceae*, accounting for 13.04% of the total sequence. The seven taxa described above as having more than 1% of the total sequence were identified at the genus level: *Treponema*, *Ruminococcus*, *Prevotella*, *Oscillospira*, *Lactobacillus*, *Coprococcus*, *Clostridium*, and *Bacteroides*. These genera accounted for 19.51% of the total sequence. Three taxa were identified at the order level accounting for 24.32% of the total sequence. In addition, the order *Clostridiales* accounting for 13.3% of the total sequence was also dominant in the fecal microbiota of sows. All other phyla with less than 1% of the total sequence accounted for 19.79% of the total sequence.

### 3.2. Microbial Distribution of Sows in the Perinatal Period

There was a significant difference in microbial changes before and after birth after the log2Fold change of phyloseq in R was used to perform the differential analysis of perinatal intestinal microbes (*p* < 0.05, Figure 3A,B). Some bacteria OTUs such as *Clostridium*, *Coprococcus*, *Streptococcus*, *Lactobacillus*, *Treponema*, *Oscillospira*, *Ruminococcus*, *Prevotella*, *Bacteroides*, *Parabacteroides*, and *Propionicimonas* were present both before and after birth. However, some bacteria were higher in one stage than the other. Bacteria that were higher during the prenatal period were *Epulopiscium*, *Methanobrevibacter*, *SMB53*, *Sphingomonas*, *Stenotrophomonas*, *Desulfovibrio*, *Myroides*, *YRC22*, *Phyllobacterium*, *Turicibacter*, *Phyllobacterium*, *Vibrio*, *Sarcina*, *Salinivibrio*, *Roseburia*, *Enterococcus*, *Corynebacterium*, *Campylobacter*, *Propionicimonas*, *p-75-a5*, *Proteus*, *Shewanella*, *Kaistobacter*, *Acinetobacter*, *Akkermansia*, *Azoarcus*, *Megamonas*, *Anaerovibrio*, *Jeotgalicoccus*, *Brachybacterium*, *Escherichia*, *Anaerofustis*, *Flexispira*, *Selenomonas*, *PSB-M-3*, *Citrobacter*, *L7A_E11*, and *Syntrophomonas*. On the other hand, bacteria that were higher during the postnatal period were *Coprobacillus*, *Bulleidia*, *Paludibacter*, *Bacillus*, *Sedimentibacter*, *Dialister*, *Fusobacterium*, *Actinobacillus*, *Propionibacterium*, *Peptoniphilus*, *Mobiluncus*, *Eubacterium*, *Aggregatibacter*, *Rhodococcus*, *Elusimicrobia*, *Sharpea*, *Haemophilus*, *Aerococcus*, *Brevundimonas*, *Anaerococcus*, *Facklamia*, *Butyricimonas*, *Veillonella*, *Megasphaera*, *Helcococcus*, *Garciella*, *ph2*, *Rummeliibacillus*, *Oxobacter*, *Finegoldia*, *Sporanaerobacter*, *Succiniclasticum*, *Mitsuokella*, *Tissierella*, *Soehngenia*, *Catenibacterium*, *Peptostreptococcus*, *Pediococcus*, and *Acidaminococcus*.

Despite results suggesting significant differences before and after birth, sow fecal microbiota was also influenced by the specific time before or after birth (Figure 1C,D). Therefore, we also determined the biomarkers of the prenatal and postnatal period using LEfSe analysis (Figure 3C,D). The results suggested that *Bacteroidales*, *Bacteroidia*, and *Prevotella* were enriched on Day 27 of the gestation period (Figure 3B) with the biomarker of *Bacteroidia* and *Bacteroidales* being Bacteroidetes. *Spirochaetes* was the only biomarker found on the 46th day of the gestation period, while *Clostridiaceae* was observed on the 64th day of the prenatal period. Data obtained after 100 days of pregnancy show that *Bacillus*, *Lactobacillales*, and *Lactobacillus* were enriched. Results of the 113th day, the last day of the test, suggested that the predominant bacteria belonged to the *Clostridium* class. Uncertain bacteria from the data were also summarized in the *Clostridia* class. Bacteria from the thick-walled *Ruminococcaceae* family also accounted for a large proportion. Results from this study suggest that two time points in early pregnancy (27th day) and late pregnancy (113th day) are the two most important time points of the bacterial species with the results being consistent with the NMDS analysis (Figure 1C).

The results of biomarker analysis after delivery showed that the postpartum period was determined at 3, 10, 14, and 21 days of lactation. Types of microbiota observed on the third day after delivery were similar to those on the 113th day of pregnancy (Figure 3C) with *Clostridiales*, *Ruminococcaceae*, *Clostridia*, and unclassified *Christensenellaceae* having the enriched proportion. *Bacteroides* and *Prevotella* biomarkers were enriched on the 10th day after delivery. *Mollicutes*, *Tenericutes*, and *RF39* were enriched on the 14th day after delivery, while *Bacilli*, *Lactobacillales*, *Veillonellaceae*, and *Mollicutes* were enriched on the 21st day after birth. The obtained results suggest that the microbiota changes constantly with the increase of the postpartum days.

### 3.3. Functional Analysis of Perinatal Microbiota

PICRUSt was used to predict the function of the microbiome during perinatal stages with results suggesting that the microbiota played similar functions during the perinatal period, though some differences existed over time (Figure 4). LEfSe analysis was also used to determine the biomarkers of microbial function in sow fecal microbiome before and after birth with results showing that the 27th, 100th, and 113th day of the prenatal period were very important for distinguishing the time point effects (Figure 4A). Fold sorting and degradation, phenylalanine tyrosine and tryptophan biosynthesis, glycosyltransferases, folate biosynthesis, histidine metabolism, and valine, leucine, and isoleucine biosynthesis were significantly prominent on Day 27 of the prenatal period. Replication and repair, DNA repair and recombination proteins, terpenoid backbone biosynthesis, primary bile acid biosynthesis, and secondary bile acid biosynthesis were enriched on the 100th day. Only transcription factors, protein kinases, and nitrotoluene degradation were enriched on the 113th day of the prenatal period, which was the last day of the critical prenatal time points.

The results of the postpartum period suggested that the biomarkers of microbial function were found on five time points with only Day 7 lacking biomarkers (Figure 4B). The functions of microbes three days after delivery had significant differences when compared with other days with 17 enriched functions. The second important time point during the postpartum period was on Day 14 where the enriched functions were 12. On the 21st day after delivery, the last time point of the whole perinatal analysis in this study, the enriched functions were carbohydrate metabolism, purine metabolism, phosphotransferase system (PTS), other ion coupled transporters, and signaling molecules and interactions. There were a few different functions on Days 5 and 10 postpartum where methane metabolism was the enriched function on the fifth day, while caffeine metabolism and base excision repair were enriched on the 10th day. It is worth noting that methane is a greenhouse gas with a higher greenhouse effect than CO_2_, and it is also a form of energy loss from feed intake. In summary, the fecal microbiota functions during the postpartum period were clearly divided into the early stage (three days) and the late stage (14 and 21 days), which was consistent with the NMDS analysis (Figure 1D).

### 3.4. Investigation of Significant Differences in OTUs Using Machine Learning

All the 16S rRNA data were analyzed using the random forest algorithm to determine the most important OTUs during the perinatal period. The results identified 29 important OTUs, which are listed in Figure 5A. The bacteria were classified into the Actinobacteria, Bacteroidetes, Firmicutes, Proteobacteria, and Spirochaetes categories. *Corynebacterium*, *Collinsella*, and *Coriobacteriaceae* were the three taxa related to Actinobacteria in the whole perinatal period with the proportion of *Coriobacteriaceae* being the highest. Five species including *Bacteroidales*, *S24-7*, *Bacteroides*, *Parabacteroides*, and *RF16* were related to Bacteroidetes during the perinatal period. Firmicutes were the most abundant during the perinatal period with several classes of genera including *Lachnospiraceae*, *Clostridiaceae*, *Clostridiales*, *Turicibacter*, *SMB53*, *Sarcina*, *Mitsuokella*, *Megasphaera*, *Dehalobacterium*, *Catenibacterium*, *Acidaminococcus*, and *Eubacterium*. Proteobacteria contains six species of genera *GMD14H09*, *Sutterella*, *Stenotrophomonas*, *Phyllobacterium*, *Haemophilus*, and *Desulfovibrio*. Heatmap results suggested that the genera were significantly shaped by prenatal and postnatal stages (Figure 5B). *Mitsuokella*, *Acidaminococcus*, *Collinsella*, *Catenibacterium*, *Eubacterium*, *Sutterella*, *Megasphaera*, *Bacteroides*, *Sphaerochaeta*, *Parabacteroides*, and *Treponema* genera and the RF16 family were higher during the postnatal period than during the prenatal period. In addition, *Megasphaera*, *Mitsuokella*, *Acidaminococcus*, and *Catenibacterium* genera were highest on the 21st day after birth. On the other hand, genera *Sarcina*, *Phyllobacterium*, *Stenotrophomonas*, *SMB53*, *Corynebacterium*, *Turicibacter*, and *Desulfovibrio* and the *Clostridiaceae* family were enriched during the prenatal period. However, there were no significant differences with their regularity during the different days of the prenatal period. The *Dehalobacterium* genus, *Lachnospiraceae* and *S24-7* families, and *Clostridiales* and *Bacteroidales* orders were sparsely distributed during the perinatal period.

Predictable functions by the machine learning analysis of 16S rRNA data using PICRUSt were ranked according to importance from the highest to the lowest (Figure 6A). Lysine biosynthesis, cysteine and methionine metabolism, benzoate degradation, sulfur metabolism, arginine and proline metabolism, cytoskeleton proteins, aminobenzoate degradation, tetracycline biosynthesis, photosynthesis-antenna proteins, apoptosis, G protein-coupled receptors, and the calcium signaling pathway were higher during the prenatal period (Figure 6B). The citrate cycle (TCA cycle), taurine, and hypotaurine metabolism were enriched during the postnatal period. The other functions were not significantly different between the prenatal and postnatal periods and were distributed on different days of the perinatal period.

## 4. Discussion

The gut microbiota of breeding sows directly affects the overall productivity and health of the host [4]. However, longitudinal studies investigating the systemic changes in the gut microbiota of sows throughout pregnancy and lactation are lacking. In the present study, the gut microbiota of breeding sows was significantly influenced by the prenatal and postnatal periods. *Methanobrevibacter* and *Akkermansia* were higher during pregnancy compared with the lactation period. *Methanobrevibacter* was reported to occupy 2.13% of the total sequences in the gut microbiota of sows [15]. Besides, *Methanobrevibacter* commonly uses CO_2_ and hydrogen, produced during bacterial and eukaryotic carbohydrate fermentation, to generate methane and release hydrogen pressure to maintain gut health [16]. In total, eighty-nine percent of the methanogens in the hindgut of Lantang gilts belonged to the genus *Methanobrevibacter* when fed with different dietary fibers [17]. Therefore, the fiber level in the diet of a pregnant sow controls the energy level. *Akkermansia* belongs to the phylum Verrucomicrobia and is a mucin-degrading bacterium, which was often negatively correlated with the diseases. This bacterium is a new probiotic used to improve the metabolic situation in both humans and animals [18]. In this study, *Akkermansia* was higher during pregnancy, which might improve the overall health situation of pregnant sows. Besides, ten genera belonging to the phylum Firmicutes and thirteen genera belonging to the phylum Proteobacteria were found to be higher during the pregnancy period compared with the postnatal period. *Desulfovibrio* belongs to the phylum Proteobacteria and is a sulfate-reducing bacteria, which utilizes H_2_ and sulfate to produce H_2_S [19]. *Desulfovibrio*, which belong to the phylum Proteobacteria, and *Turicibacter*, which belong to the phylum Firmicutes are often correlated with inflammation [20].

*Fusobacterium* and *Actinobacillus* were found to be significantly enriched during the postnatal period, which was consistent with a previous study findings [9]. *Fusobacterium* belonging to the Fusobacteria phylum is a normal bacteria in the gut microbiota of pig [21]. However, *Fusobacterium* is reported to closely correlate with low-grade inflammation and cancer in humans and other diseases in animals [22]. The relative abundance of *Fusobacterium* also increased when the piglet suffered from diarrhea disease [23]. These results illustrated that the sows might exhibit some inflammation during the postnatal period. Further studies should be conducted to investigate the mechanism and measures of improving inflammation by manipulating the gut microbiota in lactating sows. *Actinobacillus* is reported to be highly populated in low-weight-gain piglets compared with high-weight-gain piglets [24]. *Paludibacter* and *Butyricimonas* belong to the phylum Bacteroidetes and are propionate and butyric acid-producing bacterium, respectively [25,26]. These short fatty acids are important to maintain host and sow health [27]. Twenty-seven genera belonging to the phylum Firmicutes were found to be in significantly high proportions during the postnatal period. The most abundant genera were *Megasphaera*, *Succiniclasticum*, *Acidaminococcus*, and *Rummeliibacillus. Megasphaera* is a medium-chain carboxylic acid-producing bacterium [28]. *Succiniclasticum* is a propionate-producing bacterium [29]. *Acidaminococcus* is an anaerobic Gram-negative diplococcus using amino acids to produce acetate and butyrate (Rogosa et al., 1969). *Rummeliibacillus* is reported to be one of the most abundant genera in pigs [30].

To determine the bacteria differences during pregnancy and the lactation period, respectively, we used the LEfSe analysis method to investigate the biomarkers at different times of pregnancy and lactation. The biomarkers of G27 were bacteria in the phylum Bacteroidetes. *Prevotella* was the main genus of Bacteroidetes found in the gut of sows [9,15]. A previous study also reported that *Prevotella* had the highest relative abundance at the day 14 during lactation period [9]. The present study found that *Prevotella* had no significant difference between pregnancy and lactation times, with OTUs distributed equally at two time points (Figure 3A). Functional analysis revealed that amino acid metabolism was enriched at G27, including phenylalanine, tyrosine, and tryptophan biosynthesis, histidine metabolism, and valine, leucine, and isoleucine biosynthesis. *Lactobacillus* enriched at G100 might be associated with bile acid biosynthesis (Figure 4B). *Lactobacillus* was positively correlated with the bile salt hydrolase activity [31]. Bile acid is important in maintaining gut microbiota balance and host health [20]. The increase in Lactobacillus and bile acid biosynthesis in the late stage of sow pregnancy might be beneficial to the health of the sow. The biomarkers of G113 belonged to the *Clostridium* class and family *Ruminococcaceae*. Functional enrichment of substrate metabolism was not observed at the G113 time point (Figure 4B). These difference might be also affected by the cloprostenol treatment. The family *Ruminococcaceae* was reported to be significantly higher at G109 than that at the G30, L3, and L14 time points in a previous study [9]. *Ruminococcaceae* belongs to the class *Clostridia* bacteria and encodes a large number of fiber-degrading enzymes to produce SCFAs, which is highly connected to a high fiber diet in late pregnancy in sows [32]. However, this shift was caused by the change in diet, and the transition between phases needs to be further confirmed.

For the postnatal period, the marker bacteria at the L3 time point were similar to those at G113. However, more functions were enriched at L3 during the postnatal period. These functions included amino acid metabolism, adaptation, and signal transmission in bacteria. On the 10th day after delivery, the biomarkers of the diet were *Bacteroides* and *Prevotella*. On the 14th day after delivery, the biomarkers of the diet were mainly represented by the phylum Tenericutes, with the order *Mollicutes* and genus *RF39* mainly dominant. On the 21st day after delivery, the main bacteria were those in the class *Bacilli* including order *Lactobacillales*, the family *Veillonellaceae*, and order *Mollicutes*, which belongs to the phylum Tenericutes. However, the function of the phylum Tenericutes remains unclear in the gut of sows and needs to be further investigated. At the L14 and L21 time points, carbohydrate metabolism and amino acid-related enzymes were enriched, respectively. However, the relationship between the functions and the microbiome needs further confirmation.

To further confirm the important microbiome during pregnancy to lactation, random forest was used to identify the top 29 OTUs in this study. Twenty OTUs were identified at the genus level. Seven OTUs were found to be higher during the prenatal period, including *Stenotrophomonas*, *Phyllobacterium*, *SMB53*, *Turicibacter*, *Sarcina*, *Desulfovibrio*, and *Corynebacterium*. Six OTUs were found to be higher during the postnatal period, including *Megasphaera*, *Catenibacterium*, *Acidaminococcus*, *Eubacterium*, *Haemophilus*, and *Mitsuokella*. Three genera, including *Treponema*, *Parabacteroides*, and *Bacteroides*, were found to exist both in the prenatal and postnatal periods. The most important five functions of the substrates were arginine and proline metabolism, lipoic acid metabolism, taurine, and hypotaurine metabolism, lysine biosynthesis, and cysteine and methionine metabolism. Arginine and proline metabolism, lysine biosynthesis, and cysteine and methionine metabolism were higher during the prenatal period. Lipoic acid metabolism and taurine and hypotaurine metabolism were higher during the postnatal period. In summary, different important bacteria genera and functions were mainly enriched in both the prenatal and postnatal periods. However, the limit of this study was that we collected from different animals on different days of gestation, which might include the variability of the microbiota among different individuals. In the future, the same animals from pregnancy to delivery should be selected to investigate the changes in the microbiota and include more meta-data.

## 5. Conclusions

The present study suggests that the gut microbiota of breeding sows dramatically changes throughout both the prenatal and postnatal periods. Alpha diversity is higher during the postnatal period compared with the perinatal period. Different time points in both the prenatal and postnatal periods are also associated with some unique bacteria. The phenotype prediction and function are influenced by the prenatal and postnatal stages and different time points, and the most important function is arginine and proline metabolism. Therefore, these results provide systematic data for the gut microbiota of the prenatal and postnatal periods of sow.

## Figures and Tables

**Figure 1 animals-10-02254-f001:**
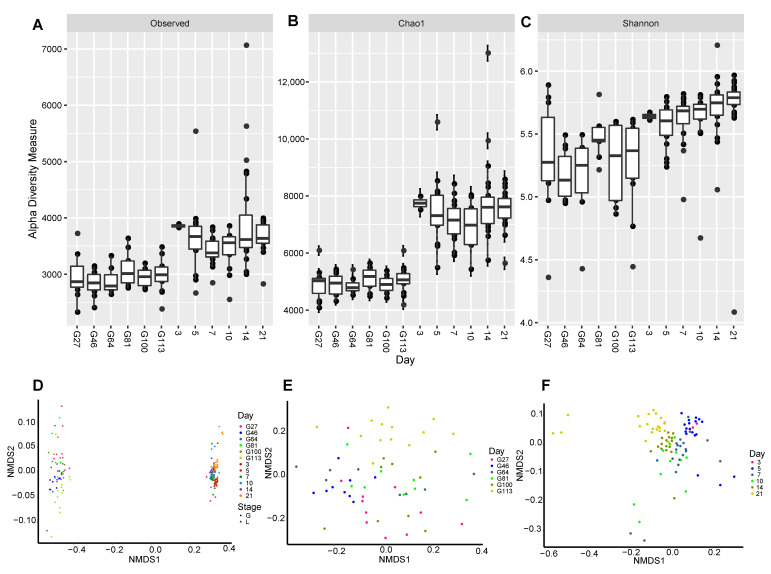
Changes in the microbiome biodiversity in the perinatal period. (**A**) Bacterial alpha diversity determined by the observed OTU index. (**B**) Bacterial alpha diversity determined by the Chao index. (**C**) Bacterial alpha diversity determined by the Shannon index. (**D**) Non-metric multidimensional scaling (NMDS) based on Bray–Curtis for all samples. (**E**) NMDS based on Bray–Curtis for the samples before birth. (**F**) NMDS based on Bray–Curtis for the samples after delivery.

**Figure 2 animals-10-02254-f002:**
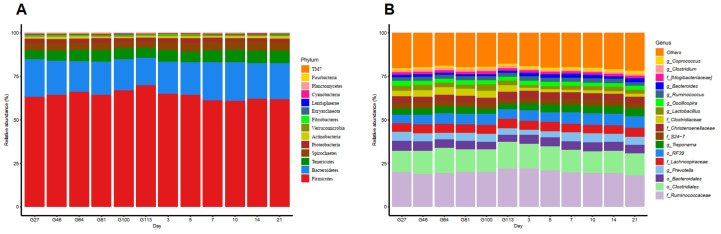
The bacterial composition in the perinatal sow feces samples determined by polymerase chain reaction amplification and 16S rRNA pyrophosphate sequencing. (**A**) The relative abundance of bacterial compositions at the phyla level of the bacteria in the feces sample. (**B**) The relative abundance of horizontal bacterial compositions at the genera level in fecal samples.

**Figure 3 animals-10-02254-f003:**
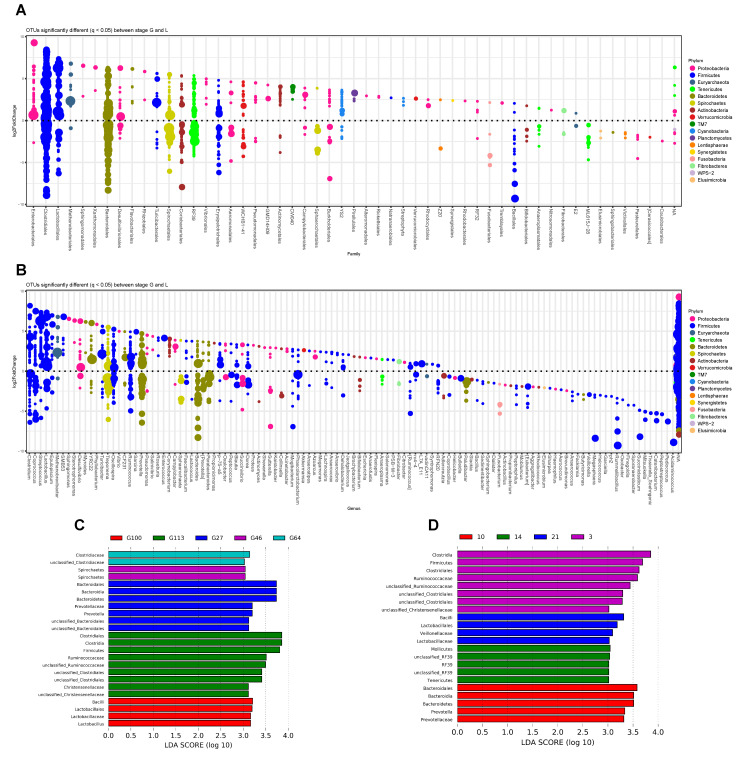
Different bacteria compositions in the perinatal period of sows. (**A**) The difference in bacteria composition between the prenatal and post periods at the family level; (**B**) the difference in bacteria composition between the prenatal and post periods at the genus level; (**C**) the differences in bacteria composition at different time points during the prenatal period; (**D**) the differences in bacteria composition at different time points during the postnatal period.

**Figure 4 animals-10-02254-f004:**
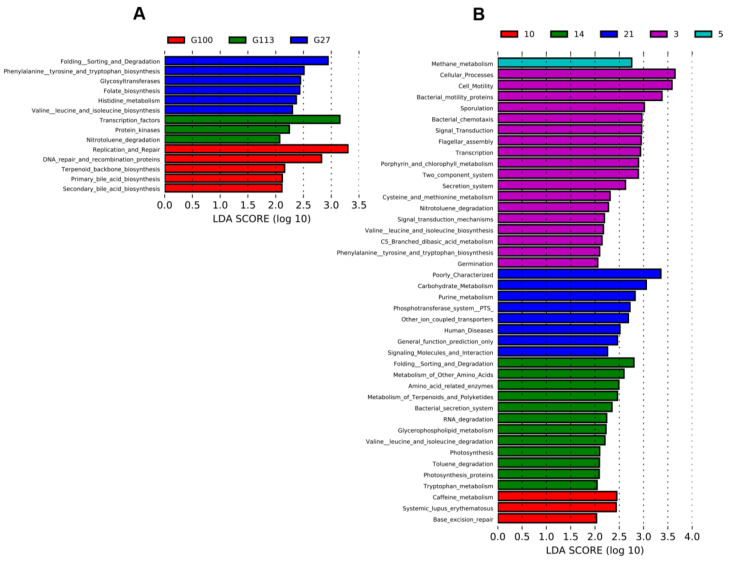
Perinatal bacterial function prediction. (**A**) The functional differences of bacteria at different time points during the prenatal period; (**B**) the functional differences of bacteria at different time points during the postnatal period.

**Figure 5 animals-10-02254-f005:**
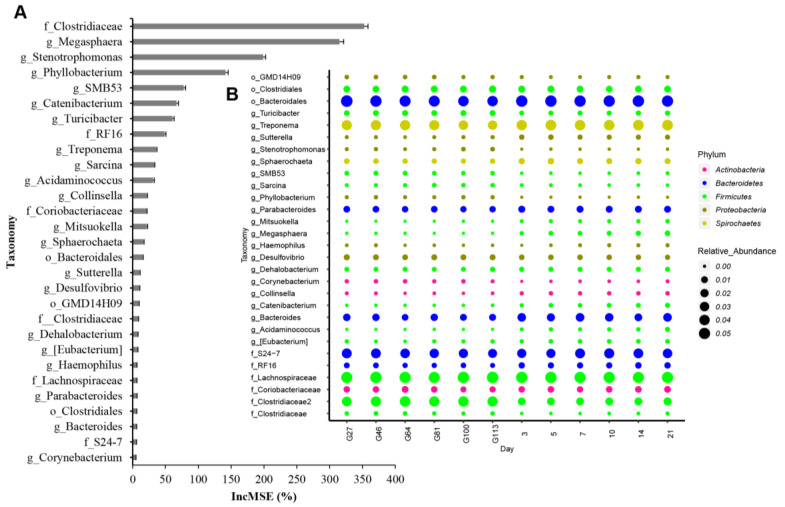
The important bacteria are identified by the machine learning algorithm random forest. (**A**) The top 29 bacteria during the perinatal period; (**B**) the distribution of the top 29 bacteria at different time points. IncMSE means increased in mean squared error.

**Figure 6 animals-10-02254-f006:**
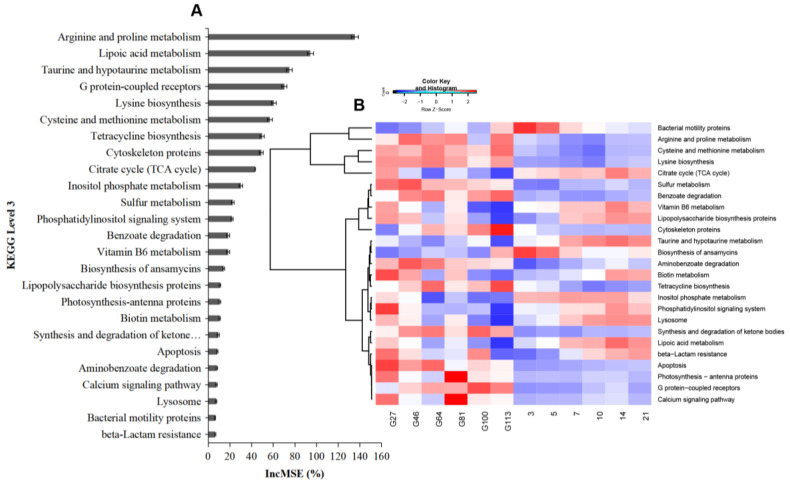
The important bacteria functions identified by the machine learning method random forest. (**A**) The top 25 bacteria functions during the perinatal period; (**B**) the distribution of the functions of the top 25 bacteria at different time points.

## Data Availability

The sequencing data in this study were deposited in the European Nucleotide Archive (ENA) under Accession Number PRJEB39955.

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
