# Peer review of "The Dynamic Changes of Gut Microbiota during the Perinatal Period in Sows"

_animals, 2020, doi:10.3390/ani10122254_

Round 1
Reviewer 1 Report
The manuscript is very interesting, studding the dynamic changes of gut microbiota during the perinatal period in sow. The theme is very important and for sure will be of interest of many scientist and pig farmers. My major concern is how the sow were selected for the study. It is well known that there is a significant variability of the microbiota among the individuals in the same herd. So, the methodology chose for the time point sampling might not reflect a reality, since no individual animal was followed during the time periods evaluated. One can understand that follow the same animal from the pregnancy to the delivery could be challenge. Thus, this fact should be addressed on the text that it may be a limitation of the study.
Also, I suggest the authors revise the results section, since some sentences should be in the discussion. And as I mention in the text “The PICRUSt predict but do not confirm the function. Thus, the sentence needs to be revised using word that express suggestions since only one can only speculate the results obtained, not words like those used in the text like “indicating” should be used “suggesting”.
My major considerations are identified on the text.

Author Response
The manuscript is very interesting, studding the dynamic changes of gut microbiota during the perinatal period in sow. The theme is very important and for sure will be of interest of many scientist and pig farmers.
My major concern is how the sow were selected for the study. It is well known that there is a significant variability of the microbiota among the individuals in the same herd. So, the methodology chose for the time point sampling might not reflect a reality, since no individual animal was followed during the time periods evaluated. One can understand that follow the same animal from the pregnancy to the delivery could be challenge. Thus, this fact should be addressed on the text that it may be a limitation of the study.
√Response: Thanks a lot for your value comments. The pregnant sows at 27, 46, 64, 81, and 100 days of gestation were selected from different animal houses at the same pig farm based on weight and health condition, which was shown on lines from 98 to 102. The methodology was often used to study the longitudinal time study. However, as review’s professional comment, this methodology also existed limitation because the variability of the microbiota among different individuals. We increased the number of animals to 8 to 11 per time points to relieve the variability of different animals. We would also address this fact on the end of the discussion part on lines from 490-494.
Also, I suggest the authors revise the results section, since some sentences should be in the discussion.
√Response: Thanks a lot for your value comments. We have changed all the marked places on the pdf text and given the detail on the following response.
And as I mention in the text “The PICRUSt predict but do not confirm the function. Thus, the sentence needs to be revised using word that express suggestions since only one can only speculate the results obtained, not words like those used in the text like “indicating” should be used “suggesting”.
√Response: Thanks a lot. We have changed the word "indicated" to "suggested" thought whole manuscript using the "Track Changes" function in Microsoft Word.
My major considerations are identified on the text.
Since different animals were used to evaluate the different time point studied, and difference in his microbiota might occur I suggest change the word "indicate" for "suggest".
√Response: Thanks a lot. We have changed the word "indicated" to "suggested" thought whole manuscript using the "Track Changes" function in Microsoft Word.
Even, with a relative number of sows at days 27 and 113, those animals were not the same, and the day 113 suffer a cloprosterol treatment that might affect the microbiota. So, these facts need to be discussed in the text. At least in the discussion section.
√Response: Thanks a lot. We have added the sentences in the discussion part on lines 460 and 490-494 to address the reasons for these differences and the limit of this study.
Reference are missing, since the author mention "most studies".
√Response: Thanks a lot. We have added the reference on line 64.
The same here, which studies?
√Response: Thanks a lot. We have added the reference on line 78.
Please, revise the use of the term “Flora”. Flora is all the plant life present in a region or time, generally the naturally occurring (indigenous) native plants.
√Response: Thanks a lot for your kind comments. We have changed the term "flora" to “microbiota" thought whole manuscript using the "Track Changes" function in Microsoft Word.
So, the data was not collectd from the same sow during the pregnacy? But from different animals with different day of gestacion.
√Response: Thanks a lot for your comment. We have added the discussion about the limit of collection fecal samples from different animals during pregnancy on lines 490-494
Please, explain the sows will only start to excrete feces after three day of the delivery?
√Response: Thanks a lot for your comment. We have added the explain why the sows will only start to excreta feces after three days of the delivery on lines 119-120. The main reason because during parturition proceeding the sow decreased the feed intake and increased the energy consume.
Please, clarify in my understanding by the observed data the author cannot reach to this conclusion, only speculate.
√Response: Thanks a lot for your value comment. In order to avoid the confusion for readers, we removed this conclusion on line 179.
Reference???(line 342)
√Response: Thanks a lot for your value comment. We have added the reference on line 408.
This sentence like many other needs to be revise. The role of Akkermansia was not evaluated, so only speculative effect could be addressed on the text.
√Response: Thanks a lot for your value comment. We have revise the sentence “In this study, Akkermansia was higher during pregnancy, which improved the overall health situation in pregnant sows.” to “In this study, Akkermansia was higher during pregnancy, which might improve the overall health situation in pregnant sows.” on line 420.

Reviewer 2 Report
The maunuscript is interesting in showing significant changes in gut microbiota and functions over a relatively short time period between peri- and postnatal periods in pigs.
However, I do have some comments/concerns.
Line 69-72: Sentences regarding effects of inulin are irrelevant to this paper.
Figures 3 & 4: The resolution of both these figures is of very poor quality to the point of being illegible.
It would be very useful and reader-friendly to have the lists of bacteria & functions as written in Lines 212-229, 264-270, 274-287, 304-322 & 328-333 to be presented in tables rather than long lists in the text.
Lines 300-304: There is no need to list all the bacteria in the text, just refer to Fig. 5.
Lines 30-33, 341-360 & 427-429: The bacterial groups mentioned maybe enriched within each time period compared to the other, but in no way can they be described as dominant if their abundance is only a fraction of the total microbiota.
What are the relative abundances of these groups? Their abundance may differ significantly at different time points but are they abundant enough to be of biological significance? There is a world of difference between a statistical change between different time points and that change being of significance.
Do any of the most abundant OTUs/bacterial taxa differ between time points?
I think the conclusion is too long and rather over-reaches on the significance of changes seen in low abundance taxa. The changes in bacterial functions seen at different time points however is interesting and this should be focused on more in the conclusions.
Author Response
The maunuscript is interesting in showing significant changes in gut microbiota and functions over a relatively short time period between peri- and postnatal periods in pigs.
However, I do have some comments/concerns.
Line 69-72: Sentences regarding effects of inulin are irrelevant to this paper.
√Response: Thanks a lot for your value comment. These sentences come from the previous studies, and we also agree with your comments that the effects of inulin are irrelevant to this paper. However, these studies were conducted about the gut microbiota of sows. Thus, we referenced this result in our paper.
Figures 3 & 4: The resolution of both these figures is of very poor quality to the point of being illegible.
√Response: Thanks a lot for your value comment. We have provided the high quality of the figures 3 and 4 during the first submit. We will ask the editor to help us again to replace the poor quality ones. Thanks a lot.
It would be very useful and reader-friendly to have the lists of bacteria & functions as written in Lines 212-229, 264-270, 274-287, 304-322 & 328-333 to be presented in tables rather than long lists in the text.
√Response: Thanks a lot for your value comment. We have shorted or deleted the lists of bacteria & functions and just listed the most important ones or just listed the total number on lines 311-312, 354. Because these bacteria and functions had been shown in the figures, if we presented in tables again that would be repeat between figures and tables. Thus, we deleted some lists of bacteria & functions as you mentioned. However, we also keep some with the summary reports for the results for others.
Lines 300-304: There is no need to list all the bacteria in the text, just refer to Fig. 5.
√Response: Thanks a lot for your value comment. We have deleted the list of all bacteria and just refer to Fig. 5 according to your comment on line 354.
Lines 30-33, 341-360 & 427-429: The bacterial groups mentioned maybe enriched within each time period compared to the other, but in no way can they be described as dominant if their abundance is only a fraction of the total microbiota. What are the relative abundances of these groups?
√Response: Thanks a lot for your value comment. We agree with your comments, the bacterial groups should be described using “enriched” terms. We have change the description throught paper. And we also checked the part from 341-360, this part only discussion about which bacteria is different between different time points and their potential functions. And also we check the average relative abundance of Methanobrevibacter, Desulfovibrio,Akkermansia, Turicibacter, Eubacterium, Actinobacillus, Paludibacter, Butyricimonas, Megasphaera, Succiniclasticum, Acidaminococcus, and Rummeliibacillus were 0.4%, 0.5%, 0.1%, 0.4%, 0.1%, 0.1%, 0.5%, 0.1%, 0.1%, 0.1%, 0.1%, and 0.1%. Because the significant OTUs identified we used was log2change method in phyloseq. We just picked out the unique appearance between between the prenatal and post periods. Thus, these unique ones are most not the dominant ones.
Their abundance may differ significantly at different time points but are they abundant enough to be of biological significance? There is a world of difference between a statistical change between different time points and that change being of significance.Do any of the most abundant OTUs/bacterial taxa differ between time points?
√Response: Thanks a lot for your value comment. These bacteria was different at different time points. However, their biological functions might not directly correlated with abundance. Whether these different bacteria play important roles on biological functions might be needed further studies, which is out scope of this study. We used random-forest to identify the most important OTUs during the whole time, the average relative abundance above 1% were f_Lachnospiraceae, o_Bacteroidales, g_Treponema, f_S24-7, f_Clostridiaceae2, g_Bacteroides, and o_Clostridiales with 5%, 5%, 4%, 3%, 3%, 2% ad 1%, respectively. Moreover, the different microbiota between different time points indentified by lefse also have dominant ones and we also lised these result on the abstract on lines 33-37.
I think the conclusion is too long and rather over-reaches on the significance of changes seen in low abundance taxa. The changes in bacterial functions seen at different time points however is interesting and this should be focused on more in the conclusions.
√Response: Thanks a lot for your value comment. We have shorted the conclusion and deleted the description about the significance of changes in low abundance taxa. And we also added the most important biomarker of functions in the conclusion and abstract. Please find the details on lines 496-502.
